# Evaluation of Prolonged Endometrial Inflammation Associated with the Periparturient Metabolic State in Dairy Cows

**DOI:** 10.3390/ani12233401

**Published:** 2022-12-02

**Authors:** Rena Sato, Saku Koyama, Miya Yasukawa, Takuma Inoue, Tomochika Matsumura, Asuka Kanazawa, Yukari Nozue, Yuriko Wada, Itaru Yoshimura, Yujiro Hagita, Hisashi Mizutani, Tsuyoshi Tajima, Tadaharu Ajito, Ryotaro Miura

**Affiliations:** 1School of Veterinary, Faculty of Veterinary Science, Nippon Veterinary and Life Science University, Kyonan, Musashino, Tokyo 180-8602, Japan; 2Fuji Animal Farm, Nippon Veterinary and Life Science University, Fujikawaguchiko, Yamanashi 401-0338, Japan

**Keywords:** endometrium, inflammation, body fat, muscle breakdown

## Abstract

**Simple Summary:**

The appearance of endometrial polymorphonuclear cells (PMN) was sequentially observed after calving by endometrial cytology, and the relationship between endometrial inflammation and metabolic state during the peripartum period was evaluated. The endometrial inflammation threshold was determined as 5.0% PMN according to PMN dynamics revealed by endometrial cytology. The present study demonstrated that cows with low body condition scores and backfat thickness had prolonged endometrial inflammation; moreover, higher blood 3-methylhistidine after calving was observed in cows with prolonged endometrial inflammation, indicating that muscle breakdown was promoted in those cows. Hence, it was proposed that evaluating body fat reservation and muscle breakdown was crucial for understanding endometrial inflammation convergence in lactating dairy cows.

**Abstract:**

The objectives of this study were to assess the sequential dynamics of the endometrial polymorphonuclear cells (PMN) after calving by endometrial cytology, and clarify the factors that cause prolonged endometrial inflammation in lactating dairy cows. A total of 33 lactating Holstein dairy cows were used from −4 to 8 wk relative to calving (0 wk: the calving week). Endometrial samples were obtained sequentially from 2 to 8 wk. Body condition score and backfat thickness were obtained weekly from −4 to 8 wk. Blood samples collected from −4 to 8 wk were analyzed for indicators of energy status, hepatic function, systemic inflammation, and calcium. Blood amino acids were measured at 2 wk. Daily milk production was determined between 5 and 65 d postpartum. Based on the sequential cytological analysis, the endometrial inflammation threshold was set at ≥5.0% PMN, and the median wk of PMN% lower than 5.0% was 4.5 wk in this study; therefore, we classified the cows into the early group (cows with endometrial inflammation converged within 4 wk: *n* = 17) and the late group (cows with endometrial inflammation converged at or after 5 wk: *n* = 16). There were no differences in daily milk production, energy status, hepatic function, blood calcium concentration, and systemic inflammatory response. The late group had lower body condition scores and backfat thickness during the experimental period, and a higher blood concentration of 3-methyl histidine, indicating muscle breakdown, was observed in the late group at 2 wk. Our findings indicated that the lack of body fat reservation during the peripartum period and the increased muscle breakdown after calving were risk factors for prolonged endometrial inflammation.

## 1. Introduction

Improvement of reproductive performance is a critical requirement in lactating dairy cows. However, postpartum endometritis, defined as endometrial inflammation, is one of the most detrimental factors for subsequent reproductive performance [1,2,3].

Numerous studies have reported endometritis effects on subsequent fertility [4,5,6]. However, the cytology day and threshold for polymorphonuclear cell (PMN) ratio (PMN%) differ among the studies, and most of the previous studies have assessed PMN% only 1 or 2 time points postpartum [5,7,8]. Although there is one report that weekly evaluate PMN% from 2 to 7 wk postpartum [9], endometrial cytology took time and it was difficult to evaluate the endometrial inflammation sequentially during the postpartum period under clinical site; therefore, the healing process of endometrial inflammation is not well evaluated.

Some studies have investigated serum metabolic patterns of endometritis before and after calving [5,8,10]. Bogado Pascottini and LeBlanc reported serum markers for healthy, purulent vaginal discharge (PVD), and subclinical endometritis cows from 7 d prepartum to 35 d postpartum [8]. In this report, subclinical endometritis had lower serum total calcium (tCa) at 7 d before calving, and greater serum non-esterified fatty acid (NEFA), β-hydroxybutyrate (BHB), globulin (GLB), and haptoglobin (Hp) in the early postpartum period. Additionally, Galvão et al. showed that cows with subclinical endometritis have higher blood NEFA and BHB, and lower intracellular PMN glycogen than healthy cows during the postpartum period [10]. In addition, Burke et al. showed that the cows with endometritis have lower albumin (ALB) and higher aspartate aminotransferase (AST) and glutamate dehydrogenase (GGT) than cows without endometritis [5]. Accordingly, endometritis has been considered associated with nutritional status, such as negative energy balance (NEB) or hepatic dysfunction, during the peripartum period and with systemic inflammation after calving. However, in the previous reports, metabolic markers for uterine diseases have been evaluated based on a single evaluation of endometrial cytology after calving. Thus, the relationship between metabolic status and sequential changes in the endometrial PMN% over the postpartum period remains unclear.

Furthermore, dairy cows undergo metabolic changes due to NEB from late gestation to early lactation and adapt to NEB by mobilization of body fat [11,12,13] and muscle protein [14]. Protein mobilization is monitored by blood 3-methylhistidine (3-MH), indicating muscle breakdown [15,16]. In addition, an increase in plasma 3-MH and a decrease in longissimus muscle thickness occurred around calving, indicating increased protein mobilization [14]. However, associations between postpartum inflammation of the endometrium and muscle breakdown remain unclarified.

The objectives of this study were (1) to clarify the dynamics of the PMN% by sequential endometrial cytology examination, and (2) to clarify the factors that cause prolonged endometrial inflammation after calving.

## 2. Materials and Methods

### 2.1. Animals, Housing, and Feeding

This study was conducted between October 2018 and February 2021, at the Fuji Animal Farm of Nippon Veterinary and Life Science University, located in Fujikawaguchiko-machi, Yamanashi, Japan. A total of 33 Holstein dairy cows (4 primiparous and 29 multiparous, parity: 3.8 ± 1.6, average milk production until 60 d in milk: 39.7 ± 3.4 kg/d; mean ± SD) were used for this experiment. All cows were calved spontaneously during the study period with no clinical diseases (retained placenta, metritis, ketosis, hypocalcemia, and displaced abomasum). Dry cows were housed in loose barns and moved to individual calving pens 3 wk before the expected calving date. Within 5 d of calving, cows were located in a tie-stall barn, milked twice daily, and had free access to water throughout the study. Dry cows were fed 15 kg timothy hay and 3 kg concentrate (18% DM CP) during the far-off period (−60 to −22 d before the expected calving date) and 15 kg timothy hay, 1 kg alfalfa hay and 3 kg concentrate (18% DM CP) during the close-up period (−21 to −1 d before the expected calving date). Lactating cows were fed 13 kg timothy hay, 3 kg alfalfa hay, 4 to 13 kg concentrate (16% DM CP), and 2.0 to 2.5 kg beet pulp. The concentrate was provided with an automatic concentrate feeder (Max Feeder HID, Orion, Nagano, Japan). The concentrate and beet pulp were provided according to each cow’s milk yield.

### 2.2. Study Design

We visited the farm once a week every Saturday to collect samples. Cows were examined from 4 wk prepartum (−4 wk: −28 to −22 days before the expected calving date) to 8 wk postpartum (1 to 7 days after calving, regarded as 0 wk).

BCS, backfat thickness (BFT), and blood samples were collected from −4 to 8 wk. The rectal temperature and vaginal discharge were examined from 0 to 8 wk. Endometrial samples were collected from 2 to 8 wk. Figure 1 shows a schematic diagram of the experimental model.

### 2.3. BCS Measurement

We measured BCS according to the method of Ferguson et al. [17], and BCS was evaluated on a scale of 1 to 5 in 0.25 increments, with a score of 1 indicating thin and a score of 5 indicating obese.

### 2.4. BFT Measurement

BFT was measured using an ultrasonic diagnostic imaging system (My Lab One Alpha, Esaote) equipped with a 10.0 MHz transrectal linear probe (SV3513, Esaote). The examination area of BFT was the sacral region between the caudal one-third point from the dorsal part of the tuber ischia (pins) to the tuber coxae (hooks). The skin spots were sprayed with alcohol and a probe was applied. The probe was applied vertically to an imaginary line between the hooks and pins. The data were saved as a video, and the skin layer and subcutaneous fascia were measured as BFT (mm).

### 2.5. Rectal Temperature Measurement

Rectal temperature was measured using a mercury thermometer.

### 2.6. Blood Sampling

Blood samples were collected from coccygeal vessels into 9 mL non-heparinized serum separator evacuated tubes, 2.0 mL EDTA-2Na evacuated tubes, and 2.0 mL sodium fluoride evacuated tubes (Venoject Ⅱ vacuum blood collection tube, Terumo). Non-heparinized tubes were coagulated for 30 min at 35 °C in an incubator. EDTA-2Na and sodium fluid tubes were placed on ice immediately after collection. Non-heparinized and sodium fluid tubes were centrifuged at 2000× *g* for 15 min at room temperature for serum or plasma separation within 1 h of collection. Serum and EDTA-2Na samples were placed at 4 °C for further analysis. Plasma samples were placed at −30 °C for amino acid analysis. In addition, for measuring whole blood glucose (GLU), BHB, and ionized Ca (iCa), whole blood samples collected from coccygeal vessels were analyzed immediately.

### 2.7. Serum Biochemical Analysis

Serum samples were analyzed for NEFA, AST, GGT, ALB, total protein (TP), blood urea nitrogen (BUN), total cholesterol (T-cho), and tCa using an automatic biochemistry analyzer (BiOLiS 15i neo, Tokyo Boeki Medisys). In addition, GLB concentration was calculated from the difference between the TP and ALB concentrations, and ALB/GLB ratio (A/G) was calculated.

### 2.8. Whole Blood Analysis

GLU and BHB concentrations were measured with Precision Xceed (Precision Xceed, Abbott Japan, Tokyo, Japan), and whole blood iCa was measured with i-STAT 1 and its cartridge (i-STAT 1 analyzer, i-STAT CHEM8+ Cartridge, Abbott Japan). In addition, the EDTA-2Na whole blood samples were analyzed for red blood cell count, hemoglobin (HGB), hematocrit (HCT), and total white blood cell (WBC) count, including total count for neutrophils (NEUT) and lymphocytes (LYM) throughout the study using a hematology analyzer (pocH-100iV Diff, Sysmex).

### 2.9. Blood Hp Measurement

Serum samples obtained at 0 wk were analyzed for Hp concentrations using a commercially available sandwich enzyme-linked immunosorbent assay kit (Bovine Haptoglobin, Immunology Consultant).

### 2.10. Blood Amino Acid Measurement

Plasma samples obtained at 2 wk were analyzed for amino acid concentrations by liquid chromatography–mass spectrometry (LC/MS) by FUJIFILM VET Systems (Tokyo, Japan). The following amino acids were determined, Leu, Lys, Ala, Arg, Asn, Asp, Gln, Glu, Gly, His, Met, Orn, Pro, Ser, Thr, Val, Ile, Phe, Trp, Tyr, 1-methylhistidine (1-MH), 3-MH, α-aminoadipic acid (α-AAA), α-aminobutyric acid (α-ABA), carnosine (Car), Cit, taurine (Tau), hydroxyproline (Hpro), sarcosine (Sar), Cys, cystathionine (Cyst), γ-amino β-hydroxy butyric acid (γ-A β-HBA), β-alanine (β-ALA), β-amino-iso-butyric acid (β-AIBA), γ-aminobutyric acid (γ-ABA), monoethanolamine (Mea), homocystine (Hcys), anserine (Ans), hydroxylysine (Hyl). The fisher ratio (BCAA/AAA) was calculated from the concentration ratio of branched-chain amino acids (BCAA; Ile, Leu, and Val) and aromatic amino acids (AAA; Tyr and Phe).

### 2.11. Evaluation of Endometrial Inflammation

Before collecting endometrial cells, we fixed the cow’s tail to the forefoot or neck with a rope, and we washed the vulva with a 500-fold diluted solution of benzalkonium chloride (Benzalkonium chloride solution, Taiyo Pharmaceutical) and wiped it with alcohol cotton. We collected endometrial samples using a cytobrush set (Metribrush, Fujihira Industry). A sterile plastic sheath (Sheath cover JEIDA, Fujihira Industry) was applied to the outer rod to protect from vaginal contamination. The cytobrush was introduced into the vagina, and the plastic sheath was pulled back when the tip of the rod reached the cervix. After passing the cervix and the tip of the rod reached the uterine body, the cytobrush was exposed from the outer rod at the uterine body and rotated onto the endometrium. The cytobrush was retracted into the rod, removed from the vagina, and then rolled onto a microscope slide. The smear was air-dried and stained using May–Grünwald–Giemsa (MGG) stain (May–Grünwald‘s eosin methylene blue solution modified for microscopy, Merck KGaA). The MGG stain specimen was observed under a microscope (OLYMPUS CX31, Olympus) at a magnification of 400×, and 200 endometrial epithelial cells, PMN, and mononuclear cells were counted to calculate the proportion of each cell. Three people observed each sample, and the average value was calculated for the proportion of each cell and used for diagnosis.

We divided the cows into the following two groups based on the median week of convergence of endometrium inflammation: the early group, in which endometrial inflammation converged within the median week and no inflammation was observed after the week point; the late group, in which endometrial inflammation converged after the median week point.

### 2.12. Vaginal Examination

Vaginal examinations were performed with a vaginal speculum from 0 to 8 wk. As with the endometrial cytology method, we fixed the cow’s tail to the forefoot or neck with a rope, and we washed the vulva. The vaginal speculum was inserted into the vagina to evaluate the presence or absence of purulent discharge in the vagina. Based on Sheldon et al. [18] classification, grade 3 or higher was determined to have PVD.

### 2.13. Daily Milk Production

Daily milk productions were recorded by a daily milk meter (MMD 500, ORION Machinery, Suzuka, Nagano, Japan). Data were analyzed from 5 to 65 d postpartum. Until 4 d after calving, most of the data were not available because of colostrum.

### 2.14. Statistical Analysis

All data analysis was performed using EZR (version 1.54, Saitama Medical Center, Jichi Medical University), a graphical user interface for R (The R Foundation for Statistical Computing). More precisely, it is a modified version of R commander designed to add statistical functions frequently used in biostatistics [19].

Change of PMN% was analyzed using the Kruskal–Wallis test to determine the main effect of the week. In addition, a multiple comparison test with Holm’s adjustment was used when a week’s effect was significant to detect significant differences among weeks.

Continuous outcomes such as BCS, BFT, rectal temperature, serum, and whole blood measurements were analyzed using two-way ANOVA with repeated measures to determine the main effects of groups (early vs. late) and week (week relative to calving date) and their interaction. Before performing ANOVA, data were subjected to the Shapiro–Wilk normality test to confirm a normal distribution and Bartlett’s test of homogeneity of variances. When a significant interaction was detected, a multiple comparison test with Holm’s adjustment was used to detect significant differences among the groups within weeks and among the weeks within groups. Amino acid, Hp, and parity data were analyzed using the Shapiro–Wilk normality test. The data were analyzed using the F-test and Student’s t-test or Welch’s test when they were significant. When the data were not normally distributed, the Mann–Whitney U test was used to analyze the data. Data from vaginal examination were compared by chi-square analysis for the proportion of PVD between groups in each week with Holm’s adjustment. A *p*-value of <0.05 indicated a significant difference, and a *p*-value of <0.1 indicated a tendency. Results are expressed as mean ± standard error of the mean (SEM) except PMN% of each week. PMN% of each week showed mean (lower quartile, upper quartile).

## 3. Results

### 3.1. PMN% Change

The PMN% change of all cows is shown in Figure 2. The PMN% of each week were 16.8% (3.1%, 40.0%) at 2 wk, 8.1% (2.6%, 22.4%) at 3 wk, 4.4% (1.7%, 7.5%) at 4 wk, 1.5% (0.7%, 4.0%) at 5 wk, 1.5% (0.4%, 3.3%) at 6 wk, 0.9% (0.6%, 1.3%) at 7 wk, and 1.3% (0.6%, 2.7%), respectively. The highest mean PMN% was observed at 2 wk and the value decreased from 2 to 5 wk. PMN% values at 5 wk (*p* = 0.004), 6 wk (*p* = 0.004), 7 wk (*p* < 0.001), and 8 wk (*p* = 0.001) were lower than the value at 2 wk. The PMN% value at 7 wk was lower than that at 3 wk (*p* = 0.005) and 4 wk (*p* = 0.006). There were no differences in PMN% among 5, 6, 7, and 8 wk. In addition, the upper quartile of PMN% from 5 to 8 wk was under 5.0%. Thus, we defined ≥5.0% PMN as a threshold for cows having endometrial inflammation. Based on this threshold, the week when endometrial inflammation converged was determined for each cow. The median wk of endometrial inflammation convergence for all cows was 4.5 wk (lower quartile: 4 wk, upper quartile: 7 wk); therefore, we defined that the early group included the cows whose endometrial inflammation converged within 4 wk and the late group included the cows whose endometrial inflammation converged on or after 5 wk. As a result, 17 cows were classified in the early group [parity 3.6 ± 0.4 (2 primiparous and 15 multiparous)], and 16 were classified in the late group [parity 4.0 ± 0.4 (2 primiparous and 14 multiparous)]. Parity was not different between the early and the late groups.

### 3.2. Daily Milk Production

Daily milk production from 5 to 65 d postpartum is shown in Figure 3. The week effect was significant (*p* < 0.001). There were no differences in the group effect and the group-by-week interaction during the study period.

### 3.3. BCS, BFT, and Rectal Temperature Analysis

The BCS, BFT, and rectal temperature measurements during the study period are shown in Figure 4, Figure 5 and Figure 6. We observed significant differences in the group effect (*p* = 0.016) and week effect (*p* < 0.001) in BCS (Figure 4). The late group had lower BCS than the early group during the study period especially postpartum. In BFT, the group effect (*p* = 0.085) indicated a tendency, and the week effect (*p* < 0.001) was significant (Figure 5). The late group tended to have lower BFT than the early group during the study period, especially prepartum. The group effect (*p* = 0.061) indicated a tendency in rectal temperature (Figure 6). The late group tended to have higher rectal temperatures than the early group during the study period. There were no differences in the group-by-week interaction between groups in BCS, BFT, and rectal temperature.

### 3.4. Detection Rate of PVD

The detection rate of PVD is shown in Figure 7. There were no differences between groups during the study period.

### 3.5. Blood Biochemical Analysis

The blood biochemical analysis results of NEFA, GLU, BHB, BUN, ALB, T-cho, AST, GGT, GLB, A/G, tCa, and iCa are shown in Figure 8, Figure 9, Figure 10, Figure 11 and Figure 12. We observed significant differences in the week’s effect in blood concentrations of NEFA (*p* < 0.001), GLU (*p* < 0.001), BHB (*p* < 0.001), BUN (*p* < 0.001), T-cho (*p* < 0.001), AST (*p* < 0.001), GGT (*p* = 0.002), GLB (*p* < 0.001), A/G (*p* < 0.001), tCa (*p* < 0.001), and iCa (*p* = 0.002). The group-by-week interaction was detected (*p* = 0.015) for GLB concentration, but multiple comparisons with Holm’s adjustment did not show any significant difference. Except for AST concentration, no differences in the group effect were observed. The early group had higher (*p* = 0.041) AST concentration during the study period especially postpartum.

### 3.6. Hematological Analysis

Blood counts of WBC, NEUT, LYM, HCT, and concentration of HGB are shown in Figure 13. The week’s effect was significant in WBC (*p* = 0.0057), NEUT (*p* < 0.001), LYM (*p* = 0.017), HCT (*p* < 0.001), and HGB (*p* < 0.001). There were no differences in the group effect and the group-by-week interaction.

### 3.7. Blood Hp

Blood concentrations of Hp are shown in Figure 14. There were no differences between groups.

### 3.8. Blood Amino Acids

Hpro, Sar, Cys, Cyst, γ-A β-HBA, β-ALA, β-AIBA, γ-ABA, Mea, Hcys, Ans, and Hyl were not detected in blood, hence we excluded these amino acids for further analysis. Blood concentrations of amino acids are shown in Table 1. We observed a significant difference between groups in 3-MH concentration (*p* = 0.049) and a tendency in 1-MH concentration (*p* = 0.090). The late group had higher 3-MH and 1-MH concentrations than the early group. There were no differences in other blood amino acid concentrations.

## 4. Discussion

We evaluated the sequential change in endometrial PMN% after calving and the relationship between the prolonged endometrial inflammation and metabolic state during the periparturient period in lactating dairy cows. The present study showed that endometrial PMN% decreased 5 wk after calving, and we set 5.0% as the threshold of endometrial inflammation existence. Cows with prolonged endometrial inflammation (converged at or after 5 wk) had lower BCS during the postpartum period and lower BFT during the prepartum and postpartum period, respectively, and higher blood 3-MH concentration at 2 wk than cows with earlier (within 4 wk) convergence of endometrial inflammation. Milk production, rectal temperature, vaginal discharge, energy status, systemic inflammation, Ca, and hematological parameters were not related to the prolonged endometrial inflammation in this study. Therefore, it was suggested that the amount of body fat reserve and muscle protein breakdown is associated with prolonged endometrial inflammation after calving.

In this study, PMN% was observed by sequential cytology. High PMN% was detected at 2, 3, and 4 wk and kept low from 5 to 8 wk. These findings showed that the endometrial inflammation converged during 4 to 5 wk. The upper quartile of PMN% at 5 wk was <5.0%; therefore, it would be reasonable to set 5.0% as the PMN threshold for the definition of endometrial inflammation in this study. Accordingly, the inflammation convergence week when PMN% decreased under 5.0% was determined in each cow, and the median of the convergence week of all cows was 4.5 wk. Thus, we classified cows whose endometrial inflammation converged within 4 wk into the early group, and cows whose endometrial inflammation converged at or after 5 wk into the late group. In previous studies, the definition of endometritis varied, for example, 18.0% during 29 to 35 d postpartum [20], 5.0% during 21 to 62 d postpartum [21], 6.0% at 42 d postpartum [5], and 8.0% at 28 to 41 d postpartum [22]. In the former study, the determination of the optimal diagnostic criteria of subclinical endometritis was 6.0% of PMN during 28 to 42 days post-partum [23]. Endometrial conditions were not evaluated sequentially in these studies; however, the PMN% threshold was similar to our study result. Therefore, we confirmed that the determination of the existence of endometritis was around 35 d in milk.

Blood 3-MH is the endpoint of muscle breakdown status [15,16]. Cows in a state of NEB experience great muscle protein and body fat mobilization [24]. Although blood 3-MH concentration was not analyzed in previous studies for uterine diseases in dairy cows, it was notable that blood 3-MH concentration at 2 wk was significantly higher in the late group. Therefore, it was indicated that increased muscle breakdown levels were related to prolonged endometrial inflammation. Conversely, we did not find significant differences in NEFA, BHB, T-cho, and GGT, indicating no difference in lipid metabolism between groups in this study.

The factor that caused promoted muscle breakdown in the late group is unclear. However, there were 2 hypotheses: (1) inflammation response is higher after calving, and (2) body fat reservation is lower in the late group.

Inflammatory cytokines induce muscle breakdown [25,26]. However, there were no differences in inflammatory markers such as GLB, A/G, WBC, NEUT, and LYM between groups throughout the periods. Besides, there was no difference in Hp concentration at 0 wk. These results indicated that inflammatory response at calving did not differ between groups. Therefore, we concluded that there is no association between the inflammation convergence timing and the inflammation degree at calving.

BCS [27] and BFT [28] are well-known indicators of fat reserves in the body of dairy cows. In the present study, BCS was lower in the late group than in the early group. This result is consistent with a previous study, showing that thin body condition before, at, and after calving is observed as one of the risk factors for subclinical endometritis [7]. In addition, the promoted muscle breakdown was observed in cows with lower BFT, and the lower BFT was detected especially before calving in the late group. In the previous study, protein mobilization is more intensive in lean cows after calving [24]. Therefore, it was considered that cows with lesser body fat before calving due to a lack of energy resources under NEB could result in promoted muscle breakdown.

It was suggested that muscle protein supplied energy, whereas there were no differences in the other amino acids except for 3-MH. Blood concentrations of some amino acids change during the postpartum period in cows with ketosis; for example, ALA concentration decreased [29], and GLY increased [30]. However, cows were not in severe NEB based on the results of NEFA and BHB concentrations in the present study; thus it was considered that blood amino acid concentrations such as ALA and GLY did not differ between groups.

Although it is unclear why the lower body fat reservation was associated with prolonged endometrial inflammation in the present study, some studies investigated the association between immune function and leptin. Leptin is an adipocytokine released from adipose tissue [31], and cows with lesser BCS have lower blood leptin concentrations [24]. In the other study, leptin is associated with T cell function, such as glucose uptake and glycolysis following T cell activation [32]. From these facts, we speculated that cows with lower fat reservations had lower leptin concentration, resulting in immune suppression and prolonged endometrial inflammation.

Fat mobilization [8,10], liver function [5,8], and serum concentration of Ca [8] are associated with uterine diseases. The outcomes of blood examination and clinical examination were compared between groups to evaluate these factors. There were no differences in NEFA, GLU, and BHB between groups, indicating that energy status between groups was not different. This result contrasts with a previous study on subclinical endometritis in cows [8,10]. In addition, we did not find differences in BUN, ALB, T-cho, GGT, and Fischer ratio between groups, indicating that protein decomposition and hepatic dysfunction did not differ between groups. Although AST tended to be higher in the early group, it was considered that the higher AST might reflect increased hepatic function. These outcomes also contrast with the results of previous studies on endometritis cows [5,8]. In addition, a previous study has reported a high Hp level or interaction in subclinical endometritis cows compared to healthy cows after calving [8]. Although rectal temperature tended to be higher in the late group than in the early group, GLB, A/G, and Hp, which are inflammation indicators, did not differ between groups and PVD detection rate. These results indicated that the inflammatory response resulting from calving did not differ between groups and was not associated with prolonged endometrial inflammation. On the other hand, although there was no significant difference in the detection rate of PVD, the detection rate was numerically higher at 7 wk; it was speculated that the prolonged endometrial inflammation is causing continued excretion of purulent discharge from the uterus. Furthermore, it has been reported that serum Ca concentration is associated with a neutrophil function [33] and uterine involution [34]; however, the blood concentrations of iCa and tCa did not affect endometrial inflammation convergence in our study. These discrepancies between previous studies and the present study may result from the difference in the classification method. For example, Burke et al. compared the cows with ≤1% uterine PMN% and >6.0% uterine PMN% categorized by quartiles based on the endometrial PMN% on 42 d after calving [5], or Bogado Pascottini and LeBlanc selected the subclinical endometritis without PVD [8]. This study focused on the PMN% of the endometrium and classified cows by the median of the timing of endometrial inflammation convergence. In addition, we did not exclude cows with PVD. These differences in classification methods caused our inconsistent results with previous reports.

In this study, subsequent reproductive performance was not investigated. Therefore, further research is needed to confirm the effect of prolonged endometrial inflammation on fertility.

## 5. Conclusions

We found that cows with prolonged endometrial inflammation had lower fat reservation during the peripartum period and promotion of muscle breakdown, evaluated by blood 3-MH concentration after calving. Alterations of lipid metabolism, hepatic function, blood Ca concentration, and systemic inflammatory response were not detected as risk factors in cows with prolonged endometrial inflammation. Further observation is needed to comprehend the mechanism of promoted muscle breakdown in cows with prolonged endometrial inflammation; however, it was proposed that the evaluation of muscle breakdown and body fat reservation during the peripartum period was crucial for the understanding of physiology and pathophysiology of endometrial inflammation convergence after calving in lactating dairy cows.

## Figures and Tables

**Figure 1 animals-12-03401-f001:**
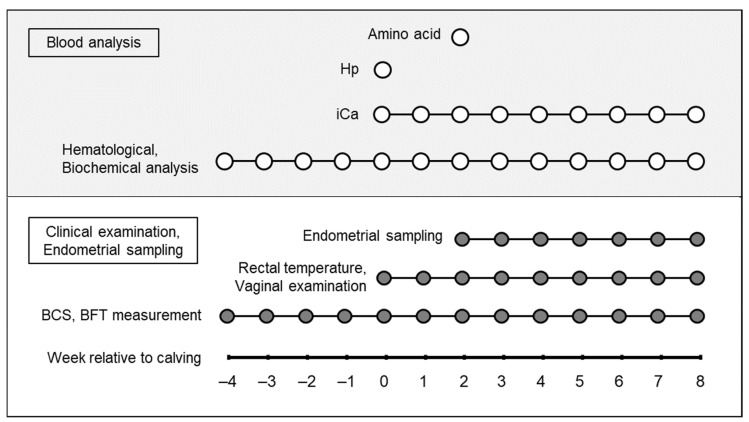
Schematic diagram of the experimented model. The week of calving is defined as 0 wk. Hp = Haptoglobin; iCa = ionized calcium; BFT = backfat thickness.

**Figure 2 animals-12-03401-f002:**
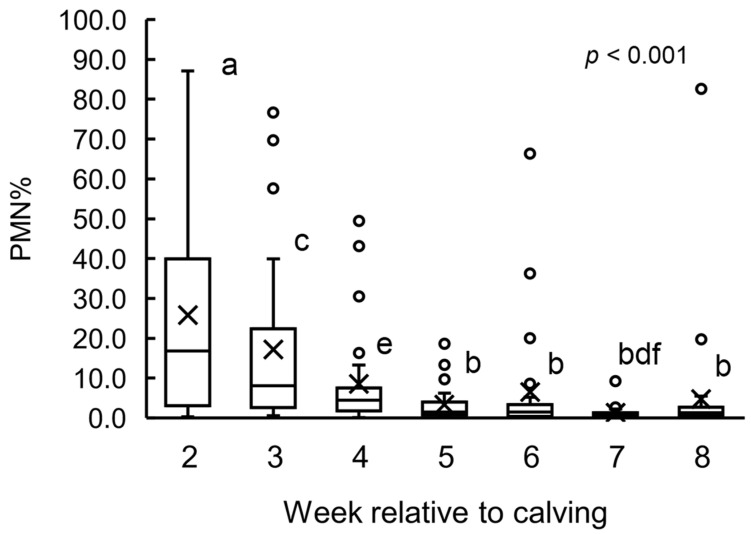
Polymorphonuclear cell (PMN) ratio (PMN%) evaluated by cytology from 2 to 8 wk postpartum. The week of calving is defined as 0 wk. Boxes represent median and interquartile range; whiskers represent minimum and maximum. Means are represented by cross. (a–b) Values at 5, 6, 7, and 8 wk are different (*p* < 0.01) than that at 2 wk. (c–d) Values with different letters are different (*p* < 0.01). (e–f) Values with different letters are different (*p* < 0.01). Means are represented by cross; outliners are represented by white circles; upper line represented upper quartile; middle line represented median; lower line represented lower quartile.

**Figure 3 animals-12-03401-f003:**
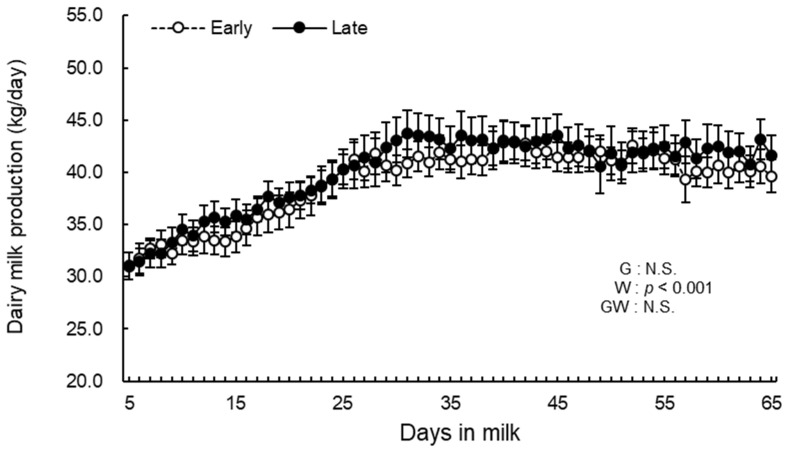
Daily milk production from 5 to 65 d postpartum for cows classified into the early group (Early) and late group (Late). Early = cows with endometrial inflammation converged within 4 wk (the week of calving is defined as 0 wk), *n* = 17; Late = cows with endometrial inflammation converged at or after 5 wk, *n* = 16. G = group effect; W = week effect; GW = group-by-week interaction. Data are shown as mean ± SEM.

**Figure 4 animals-12-03401-f004:**
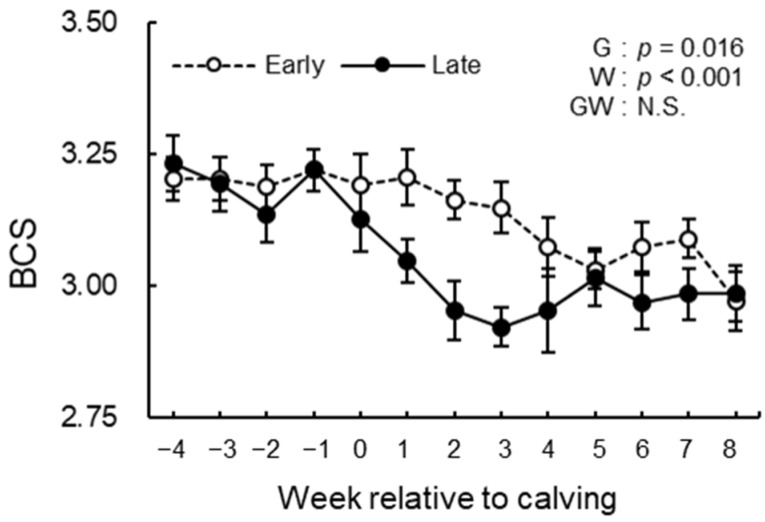
BCS from −4 to 8 wk relative to calving for cows classified into the early group (Early) and the late group (Late). Early = cows with endometrial inflammation converged within 4 wk, *n* = 17; Late = cows with endometrial inflammation converged at or after 5 wk, *n* = 16. The week of calving is defined as 0 wk. G = group effect; W = week effect; GW = group-by-week interaction. Data are shown as mean ± SEM.

**Figure 5 animals-12-03401-f005:**
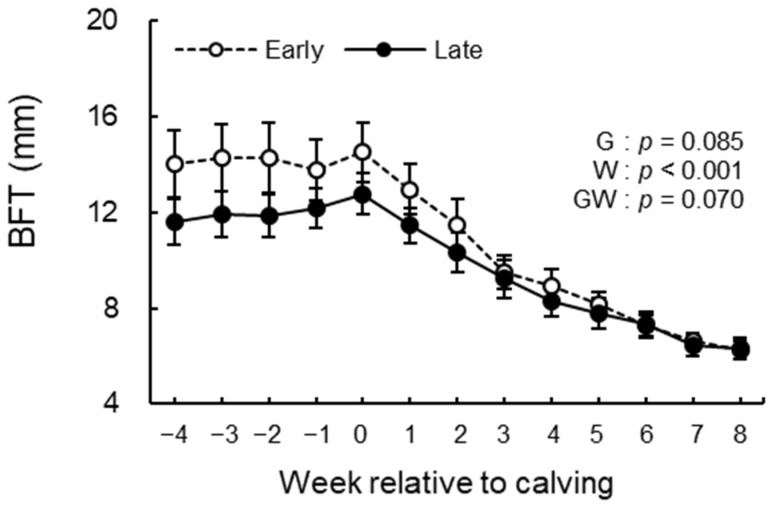
BFT from −4 to 8 wk relative to calving for cows classified into the early group (Early) and the late group (Late). Early = cows with endometrial inflammation converged within 4 wk, *n* = 17; Late = cows with endometrial inflammation converged at or after 5 wk, *n* = 16. The week of calving is defined as 0 wk. G = group effect; W = week effect; GW = group-by-week interaction. Data are shown as mean ± SEM.

**Figure 6 animals-12-03401-f006:**
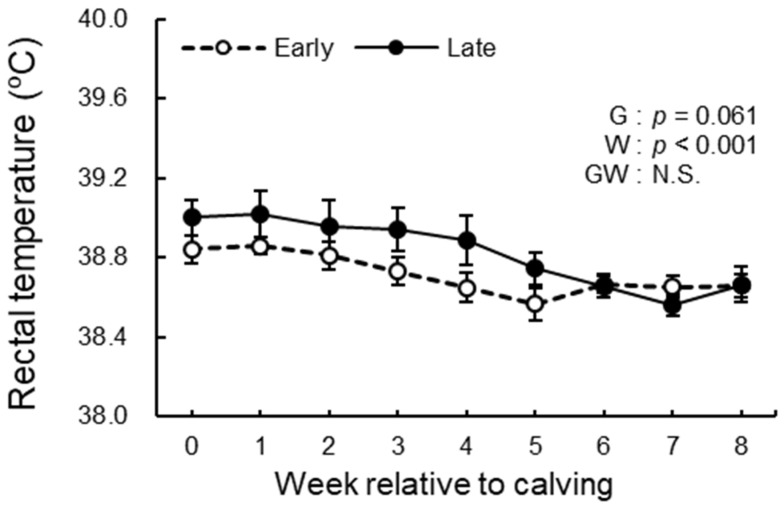
Rectal temperature from −4 to 8 wk relative to calving for cows classified into the early group (Early) and the late group (Late). Early = cows with endometrial inflammation converged within 4 wk, *n* = 17; Late = cows with endometrial inflammation converged at or after 5 wk, *n* = 16. The week of calving is defined as 0 wk. G = group effect; W = week effect; GW = group-by-week interaction. Data are shown as mean ± SEM.

**Figure 7 animals-12-03401-f007:**
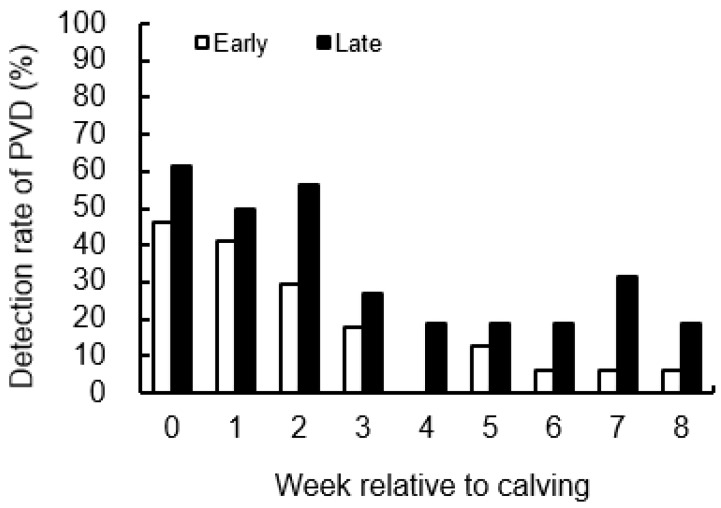
Detection rate of purulent vaginal discharge (PVD) from 0 to 8 wk relative to calving for cows classified into the early group (Early) and the late group (Late). Early = cows with endometrial inflammation converged within 4 wk, *n* = 17; Late = cows with endometrial inflammation converged at or after 5 wk, *n* = 16. The week of calving is defined as 0 wk.

**Figure 8 animals-12-03401-f008:**
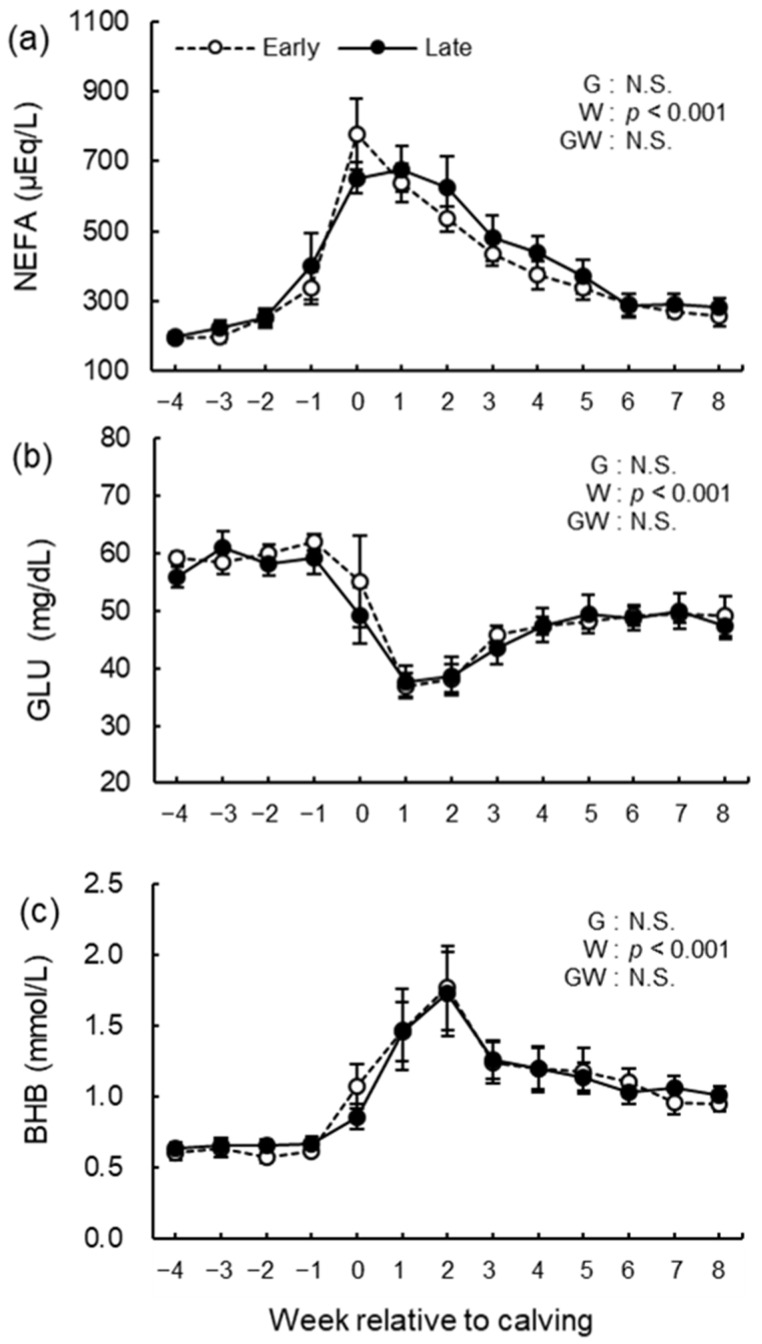
Serum concentrations of (**a**) non-esterified fatty acid (NEFA) and whole blood concentrations of (**b**) glucose (GLU), and (**c**) β-hydroxybutyrate (BHB) from −4 to 8 wk relative to calving for cows classified into the early group (Early) and the late group (Late). Early = cows with endometrial inflammation converged within 4 wk, *n* = 17; Late = cows with endometrial inflammation converged at or after 5 wk, *n* = 16. The week of calving is defined as 0 wk. G = group effect; W = week effect; GW = group-by-week interaction. Data are shown as mean ± SEM.

**Figure 9 animals-12-03401-f009:**
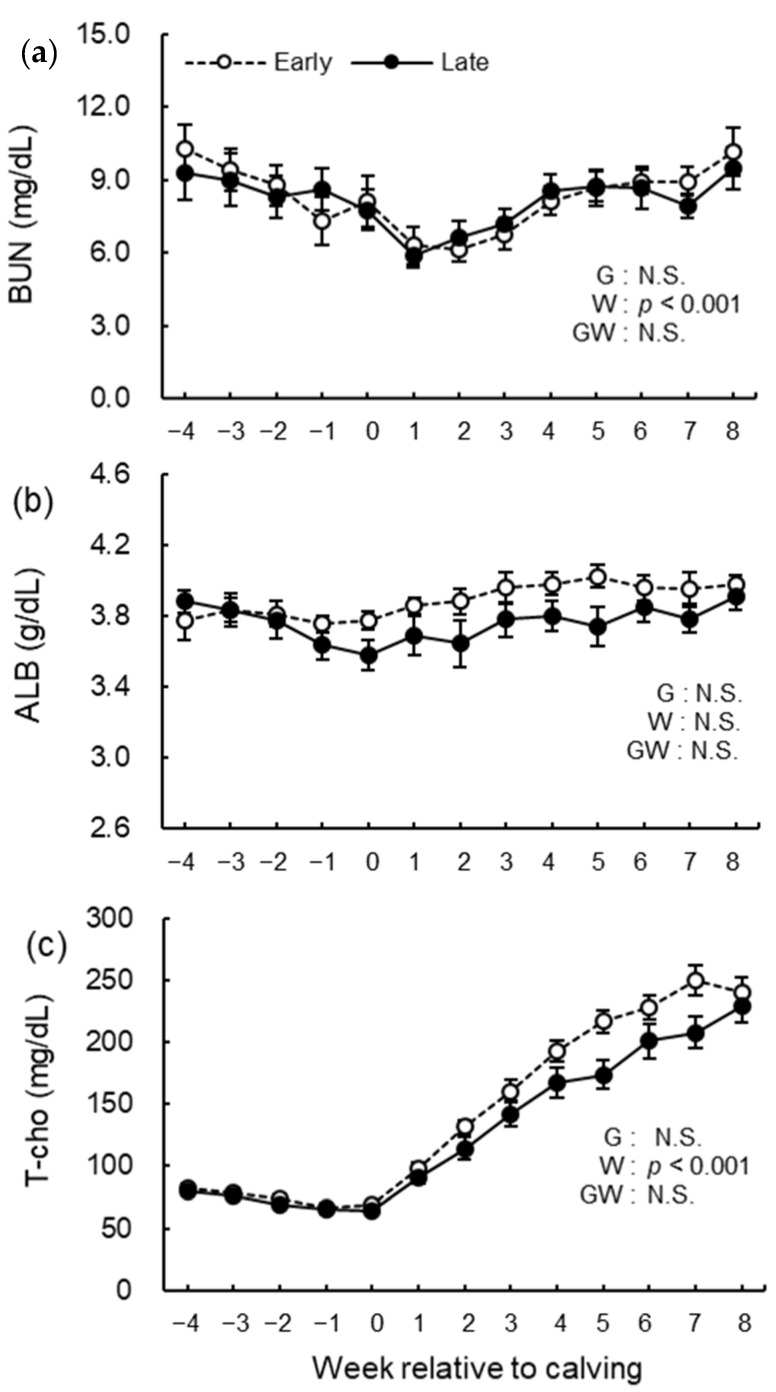
Serum concentrations of (**a**) blood urea nitrogen (BUN), (**b**) albumin (ALB), and (**c**) total cholesterol (T-cho) from −4 to 8 wk relative to calving for cows classified into the early group (Early) and the late group (Late). Early = cows with endometrial inflammation converged within 4 wk, *n* = 17; Late = cows with endometrial inflammation converged at or after 5 wk, *n* = 16. The week of calving is defined as 0 wk. G = group effect; W = week effect; GW = group-by-week interaction. Data are shown as mean ± SEM.

**Figure 10 animals-12-03401-f010:**
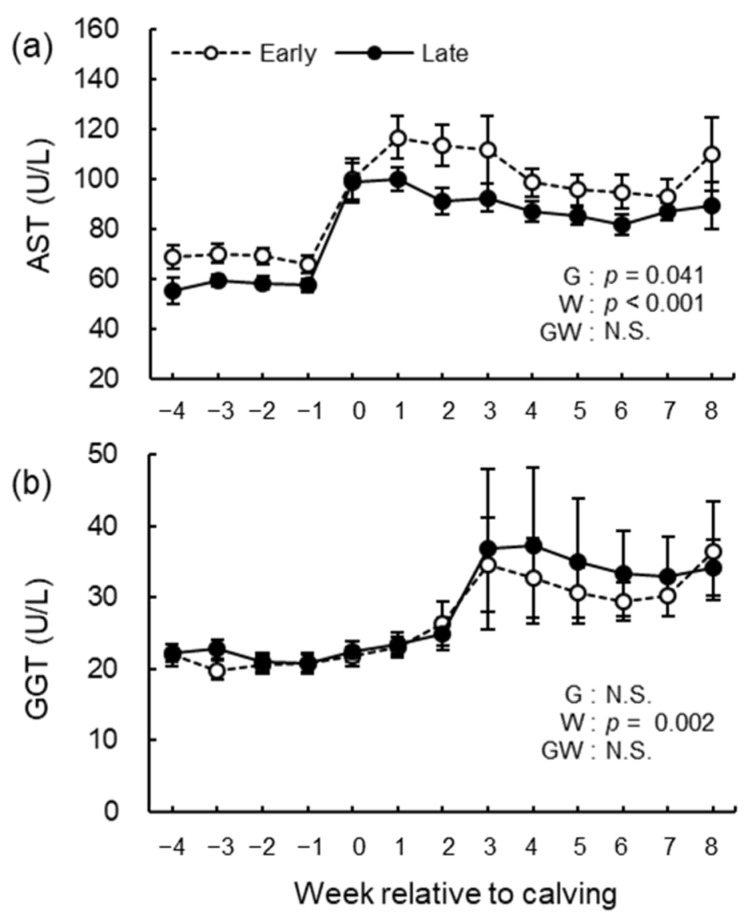
Serum concentrations of (**a**) aspartate aminotransferase (AST) and (**b**) glutamate dehydrogenase (GGT) from −4 to 8 wk relative to calving for cows classified into the early group (Early) and the late group (Late). Early = cows with endometrial inflammation converged within 4 wk, *n* = 17; Late = cows with endometrial inflammation converged at or after 5 wk, *n* = 16. The week of calving is defined as 0 wk. G = group effect; W = week effect; GW = group-by-week interaction. Data are shown as mean ± SEM.

**Figure 11 animals-12-03401-f011:**
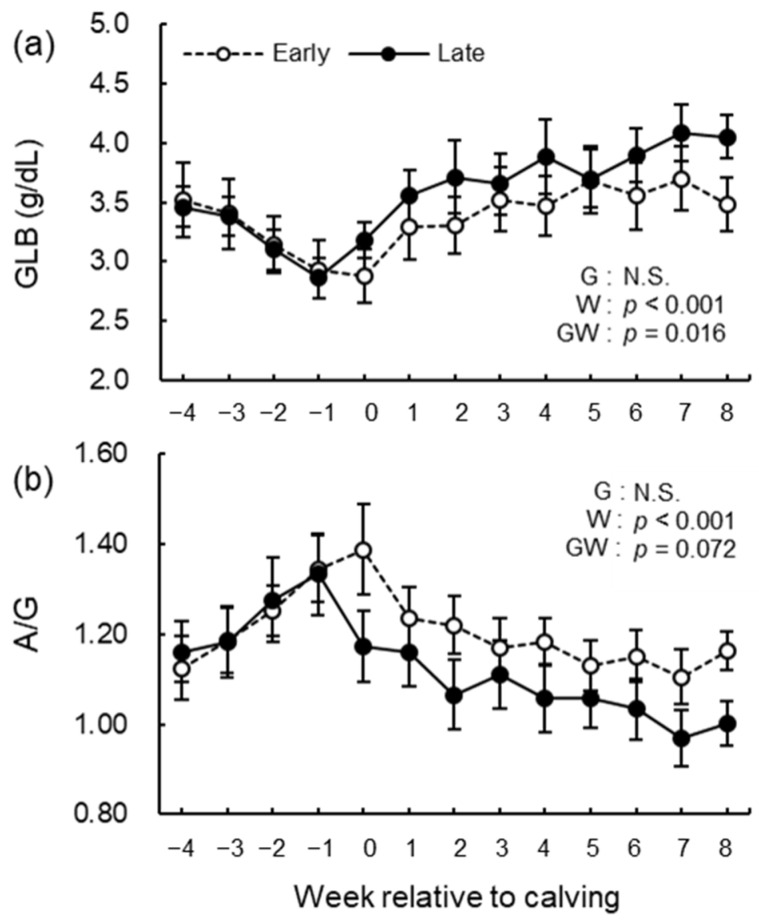
Serum concentrations of (**a**) globulin (GLB) and (**b**) albumin/GLB (A/G) from −4 to 8 wk relative to calving for cows classified into the early group (Early) and the late group (Late). Early = cows with endometrial inflammation converged within 4 wk, *n* = 17; Late = cows with endometrial inflammation converged at or after 5 wk, *n* = 16. The week of calving is defined as 0 wk. G = group effect; W = week effect; GW = group-by-week interaction. Data are shown as mean ± SEM.

**Figure 12 animals-12-03401-f012:**
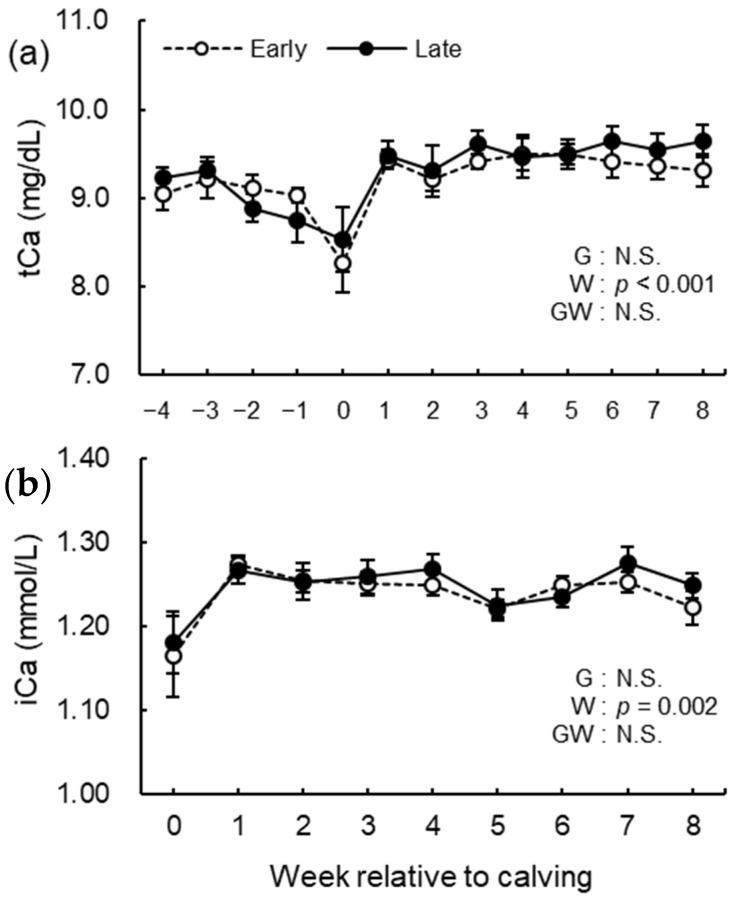
Whole blood concentration of (**a**) total calcium (iCa) from 0 to 8 wk relative to calving and serum concentrations of (**b**) ionized Ca (iCa) from −4 to 8 wk relative to calving for cows classified into the early group (Early) and the late group (Late). Early = cows with endometrial inflammation converged within 4 wk, *n* = 17; Late = cows with endometrial inflammation converged at or after 5 wk, *n* = 16. The week of calving is defined as 0 wk. G = group effect; W = week effect; GW = group-by-week interaction. Data are shown as mean ± SEM.

**Figure 13 animals-12-03401-f013:**
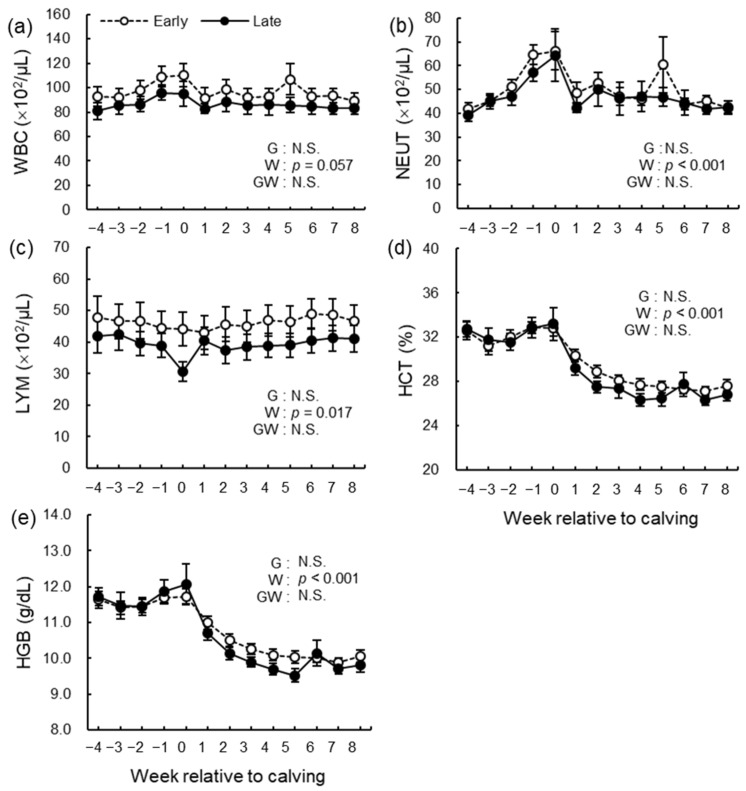
Blood counts of (**a**) white blood cell (WBC), (**b**) neutrophils (NEUT), (**c**) lymphocytes (LYM), (**d**) hematocrit (HCT), and (**e**) hemoglobin (HGB) from −4 to 8 wk relative to calving for cows classified into the early group (Early) or the late group (Late). Early = cows with endometrial inflammation converged within 4 wk, *n* = 17; Late = cows with endometrial inflammation converged at or after 5 wk, *n* = 16. The week of calving is defined as 0 wk. G = group effect; W = week effect; GW = group-by-week interaction. Data are shown as mean ± SEM.

**Figure 14 animals-12-03401-f014:**
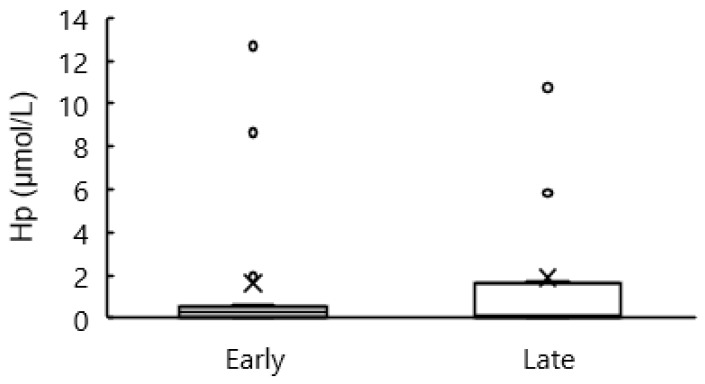
Blood haptoglobin (Hp) concentration at 0 wk relative to calving for cows classified into the early group (Early) and the late group (Late). Early = cows with endometrial inflammation converged within 4 wk, *n* = 17; Late = cows with endometrial inflammation converged at or after 5 wk, *n* = 16. The week of calving is defined as 0 wk. G = group effect; W = week effect; GW = group-by-week interaction. Means are represented by cross; outliners are represented by white circles; upper line represented upper quartile; middle line represented median; lower line represented lower quartile.

**Table 1 animals-12-03401-t001:** Differences of blood amino acid concentrations (means ± SEM; nmol/mL) between the early and the late groups at 2 wk ^1^.

Amino Acid ^2.^	Early	Late
Leu	148.7 ± 6.1	133.9 ± 7.4
Lys	73.5 ± 3.4	68.7 ± 3.5
Ala	200.6± 12.3	203.8 ± 16.6
Arg	57.1 ± 2.9	62.5 ± 7.1
Asn	39.4 ± 1.8	38.3 ± 1.8
Asp	5.5 ± 0.5	5.2 ± 0.3
Gln	194.9 ± 6.6	204.9 ± 7.3
Glu	54.5 ± 1.7	52.2 ± 2.3
Gly	448.9 ± 18.3	477.8 ± 35.3
His	46.9 ± 2.2	50.1 ± 2.6
Met	19.6 ± 1.2	18.6 ± 1.2
Orn	24.1 ± 1.4	24.2 ± 1.3
Pro	75.0 ± 2.6	74.2 ± 4.3
Ser	79.7 ± 5.1	79.0 ± 5.3
Thr	69.4 ± 5.6	62.4 ± 6.2
Val	252.9 ± 10.9	236.1 ± 12.2
Ile	157.9 ± 9.5	145.5 ± 9.4
Phe	49.5 ± 1.4	47.5 ± 1.7
Trp	35.1 ± 1.7	31.1 ± 2.3
Tyr	42.6 ± 2.0	38.7 ± 2.7
1-MH	10.4 ± 0.6 ^A^	12.3 ± 0.8 ^B^
3-MH	7.9 ± 0.6 ^a^	10.4 ± 1.0 ^b^
α-AAA	2.8 ± 0.4	2.9 ± 0.4
α-ABA	31.7 ± 2.3	33.7 ± 3.1
Car	8.9 ± 0.6	9.3 ± 1.1
Cit	59.2 ± 2.3	66.4 ± 3.8
Tau	36.2 ± 2.2	40.4 ± 4.2
Fischer ratio	5.7 ± 0.4	5.5 ± 0.5

^A,B^ Mean values in the same row with different superscripts differ (*p* < 0.1) between groups. ^a,b^ Mean values in the same row with different superscripts differ (*p* < 0.05) between groups. ^1^ The week of calving is defined as 0 wk (1 to 7 days after calving regarded as 0 wk). ^2^ Amino acids: 1-MH = 1-methylhistidine; 3-MH = 3-methylhistidine; α-AAA = α-aminoadipic acid; α-ABA = α-aminobutyric acid; Car = carnosine, Tau = taurine.

## Data Availability

The data presented in this study are available on request from the corresponding author. The data are not publicly available due to restrictions by the research group.

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
