# Peer review of "Evaluation of Prolonged Endometrial Inflammation Associated with the Periparturient Metabolic State in Dairy Cows"

_animals, 2022, doi:10.3390/ani12233401_

Round 1
Reviewer 1 Report
This is an interesting study investigating the periparturient dynamics of the relationship between endometrial inflammation and metabolic state in dairy cows. The manuscript is well written and presented.
Minor points:
Line 18 – Evaluation of muscle breakdown should be described here.
Line 52 – Please explain why the healing process of endometrial inflammation was not well evaluated.
Lines 388–399 – what about your hypothesis (Lines 79–82)?
Line 456 – change please “caused” to “was associated”
Line 457 – investigated.
Line 465 – add references.
Author Response
Reviewer 1
This is an interesting study investigating the periparturient dynamics of the relationship between endometrial inflammation and metabolic state in dairy cows. The manuscript is well written and presented.
AU:
Thank you for reviewing our manuscript and thank you for your appropriate and constructive opinion and comments. Responses to your comments are describe below. Please confirm our manuscript.
Minor points:
Line 18 – Evaluation of muscle breakdown should be described here.
AU:
Thank you for your suggestion. We described the evaluation of muscle breakdown Line 18-20.
Line 52 – Please explain why the healing process of endometrial inflammation was not well evaluated.
AU:
Thank you for your suggestion. Because the endometrial cytology take time in the clinical situation, it is difficult to evaluate the endometrial inflammation sequentially during postpartum period. Therefore, many of researches conducted one- or two-time endometrial cytology after calving to evaluated endometrial inflammation by minimize numbers of cytology for more efficient diagnosis of endometritis. We describe as below in Line 52-54.
“, endometrial cytology took time and it was difficult to evaluate the endometrial inflammation sequentially during postpartum period under clinical site; therefore, the healing process of endometrial inflammation is not well evaluated”
Lines 388–399 – what about your hypothesis (Lines 79–82)?
AU:
Yes, we intended that the hypothesis of Lines 388-399 was Lines 79-82. However, we think that we should delete Lines 388-399, because the contents of Lines 388-399 were the answer of Lines 77-78, which were the objectives of this study. Thank you for your indication.
Line 456 – change please “caused” to “was associated”
AU:
Thank you for your indication. We changed “caused” to “was associated” in Line 465.
Line 457 – investigated.
AU:
Thank you for your indication. We changed “investigate” to “investigated” in Line 466.
Line 465 – add references.
AU:
Thank you for your suggestion. We added references in Line 473.
Reviewer 2 Report
Comments.
In section 2. Materials and Methods¸2.4. BFT measurement:
It would be important for the authors to clearly explain the exact point of measurement of rump fat. It should be remembered that in the use of ultrasound for carcass quality, the P8 (place of measurement of rump fat) is measured at the intersection of the gluteus medius and biceps femoris muscles. In this way and the face of a precise anatomical description, all measurements follow a single criterion without giving rise to biased interpretations.
Author Response
Reviewer 2
Comments.
In section 2. Materials and Methods¸2.4. BFT measurement:
It would be important for the authors to clearly explain the exact point of measurement of rump fat. It should be remembered that in the use of ultrasound for carcass quality, the P8 (place of measurement of rump fat) is measured at the intersection of the gluteus medius and biceps femoris muscles. In this way and the face of a precise anatomical description, all measurements follow a single criterion without giving rise to biased interpretations.
AU:
Thank you for your suggestion. In our study, we decided that the examination area of BFT was the sacral region between the caudal one-third point from the dorsal part of the tuber ischia (pins) to the tuber coxae (hooks). We did not describe the direction of prove to the line between the hooks and pins. We changed sentences as below Lines 120-126.
“BFT was measured using an ultrasonic diagnostic imaging system (My Lab One Alpha, Esaote) equipped with 10.0 MHz transrectal linear probe (SV3513, Esaote). The examination area of BFT was the sacral region between the caudal one-third point from the dorsal part of the tuber ischia (pins) to the tuber coxae (hooks). The skin spots were sprayed with alcohol, and a probe was applied. The prove was applied vertically to an imaginary line between the hooks and pins. The data were saved as a video, and the skin layer and subcutaneous fascia was measured as BFT (mm).”
Reviewer 3 Report
Reviewer comments:
This study investigated the mechanisms of onset and convergence of post-partum endometritis (endometrial inflammation) in dairy cows, with a particular focus on prolonged endometrial inflammation, and further evaluated the relationship between the evaluation of uterine recovery by post-partum VDS and cytology and the evaluation of in vivo metabolic status by blood tests. The results of this study suggest the possibility of prolonged spontaneous recovery of endometritis by mobilization of body fat and muscle proteins in dairy cows with normal calving. However, the authors’ descriptions of the definition of endometrial inflammation, timing of examination, and the purpose of the examination are vague.
In this study, the terms “prolonged endometrial inflammation” and “persistence of endometrial inflammation” are used in the manuscript. Are these terms to be interpreted synonymously? I would encourage the uniform usage of “prolonged endometrial inflammation”, as used in the Title of this study.
Specific comments
1) Why was Hp measured only at W0? The reference (https://doi.org/10.1016/j.theriogenology.2020.06.005) reports measurement at later time points, such as 7, 14, and 35 days after parturition. If this study focuses only the inflammatory status at calving, the authors should be mentioned more closely in the introduction. However, stress, such as inflammation or infection, would certainly be present at calving; as such, measuring only at W0 does not seem sufficient to investigate Hp. If serum samples are available at other time points, the reviewer recommends that they be measured and reevaluated.
2) L223: Results of PMN% is very important in this study; please show the values for PMN% from 2 to 8 wks in the manuscript.
3)L258~261: Please add the following to the manuscript:
L258: The late group had lower BCS than the early group during the study period “especially postpartum”.
L260: The late group tended to have lower BFT than the early group during the study period “especially prepartum”.
4)Figure7: Although the difference is not significant, why did the proportion of PVD increase in W7 in the Late group? Could we consider that the prolonged endometrial inflammation is causing continued excretion of purulent discharge from the uterus?
5) L301: Please add the following to the manuscript.:
The early group had higher (P = 0.041) AST concentration during the study period “especially postpartum”.
6)L392~395: Please modify the wording as follows:
Cows with prolonged endometrial inflammation (converged at or after 5 wk) had lower BCS during the postpartum period and lower BFT during the prepartum and postpartum period, respectively.
7) L400~L406: These sentences are not required in the discussion, as they repeat information already given in the introduction.
8)L429: ”we did not find significant differences in NEFA, BHB, T-cho, AST, or GGT, indicating no difference in lipid metabolism between groups in this study”
L471: “Although AST tended to be higher in the early group, it was considered that the higher AST might reflect increased hepatic function”
These sentences are inconsistent with discussion of AST. Please show more consistency in the discussion of AST.
9)L456~L463: Is Leptin measured in this study? The reviewer could not find any mention of this. If author did not mention Leptin, please revise the manuscript and Figure 1.
Author Response
Reviewer 3
Reviewer comments:
This study investigated the mechanisms of onset and convergence of post-partum endometritis (endometrial inflammation) in dairy cows, with a particular focus on prolonged endometrial inflammation, and further evaluated the relationship between the evaluation of uterine recovery by post-partum VDS and cytology and the evaluation of in vivo metabolic status by blood tests. The results of this study suggest the possibility of prolonged spontaneous recovery of endometritis by mobilization of body fat and muscle proteins in dairy cows with normal calving. However, the authors’ descriptions of the definition of endometrial inflammation, timing of examination, and the purpose of the examination are vague.
AU:
Thank you for reviewing our manuscript and thank you for your appropriate and constructive opinion and comments. Responses to your comments are describe below. Please confirm our manuscript.
In this study, the terms “prolonged endometrial inflammation” and “persistence of endometrial inflammation” are used in the manuscript. Are these terms to be interpreted synonymously? I would encourage the uniform usage of “prolonged endometrial inflammation”, as used in the Title of this study.
AU:
Thank you for your suggestion. As you mentioned, “prolonged endometrial inflammation” and “persistence of endometrial inflammation” are same meaning. Therefore, we changed “persistence of endometrial inflammation” to “prolonged endometrial inflammation”.
Specific comments
1) Why was Hp measured only at W0? The reference (https://doi.org/10.1016/j.theriogenology.2020.06.005) reports measurement at later time points, such as 7, 14, and 35 days after parturition. If this study focuses only the inflammatory status at calving, the authors should be mentioned more closely in the introduction. However, stress, such as inflammation or infection, would certainly be present at calving; as such, measuring only at W0 does not seem sufficient to investigate Hp. If serum samples are available at other time points, the reviewer recommends that they be measured and reevaluated.
AU:
Thank you for your suggestion. And, this point is very important for discussion. In the research of Bogado Pascottini O and LeBlanc SJ (https://doi.org/10.1016/j.theriogenology.2020.06.005), blood sampling day were precisely determined at 1, 3, 5, 7, 14±1, 21±1, and 35 days after calving. However, in our study, blood sampling day had 7 days range in each week. Because we thought the blood Hp concentration would vary depending on the day from calving, we thought that it was difficult to evaluate Hp after 1 to 8 wk in our study design. However, from the result of research of Bogado Pascottini O and LeBlanc SJ, blood Hp concentrations at 3, 5, 7 days after calving were persistently higher in subclinical endometritis cow (SCE) than those of healthy cows; so, we thought that measuring blood Hp concentration within 7 days after calving (0 wk) could evaluate the condition of systemic inflammation after calving between Early and Late groups. Therefore, we focused on blood Hp concentration at 0 wk.
2) L223: Results of PMN% is very important in this study; please show the values for PMN% from 2 to 8 wks in the manuscript.
AU:
Thank you for your suggestion. We described the values of PMN% from 2 to 8wk in the Result section of 3.1. PMN% change Line 229-232.
3)L258~261: Please add the following to the manuscript:
L258: The late group had lower BCS than the early group during the study period “especially postpartum”.
L260: The late group tended to have lower BFT than the early group during the study period “especially prepartum”.
AU:
Thank you for your suggestion. We added “especially postpartum” and “especially prepartum” in each sentence Line 267 and Line 269-270.
4)Figure7: Although the difference is not significant, why did the proportion of PVD increase in W7 in the Late group? Could we consider that the prolonged endometrial inflammation is causing continued excretion of purulent discharge from the uterus?
AU:
Thank you for your suggestion and discussion. We think that reviewer’s discussion is valid to incorporate to discussion. Therefore, we added sentences as below in L489-492.
“On the other hand, although there was no significant difference in detection rate of PVD, detection rate was numerically higher at 7wk; it was speculated that the pro-longed endometrial inflammation is causing continued excretion of purulent discharge from the uterus.”
5) L301: Please add the following to the manuscript.:
The early group had higher (P = 0.041) AST concentration during the study period “especially postpartum”.
AU:
Thank you for your suggestion. We added “especially postpartum” in Line 310.
6)L392~395: Please modify the wording as follows:
Cows with prolonged endometrial inflammation (converged at or after 5 wk) had lower BCS during the postpartum period and lower BFT during the prepartum and postpartum period, respectively.
AU:
Thank you for your suggestion. We changed as you described Line 400-402.
7) L400~L406: These sentences are not required in the discussion, as they repeat information already given in the introduction.
AU:
Thank you for your suggestion. We deleted L409~L415 from this manuscript.
8)L429: ”we did not find significant differences in NEFA, BHB, T-cho, AST, or GGT, indicating no difference in lipid metabolism between groups in this study”
L471: “Although AST tended to be higher in the early group, it was considered that the higher AST might reflect increased hepatic function”
These sentences are inconsistent with discussion of AST. Please show more consistency in the discussion of AST.
AU:
Thank you for you indication. To avoid the confusing for readers, we excluded the word of “AST” from L438.
9)L456~L463: Is Leptin measured in this study? The reviewer could not find any mention of this. If author did not mention Leptin, please revise the manuscript and Figure 1.
AU:
Thank you for your indication. We are sorry for confusing you as our mistake. We did not measure Leptin in this study. We modified manuscript and Figure 1.
Round 2
Reviewer 3 Report
The reviewers' comments were appropriately and sincerely addressed, and the manuscript was well revised. In addition, the revisions pointed out by other reviewers have made it more straightforward.
I recommend that it be accepted for publication.